# Exploiting Nanomaterials for Optical Coherence Tomography and Photoacoustic Imaging in Nanodentistry

**DOI:** 10.3390/nano12030506

**Published:** 2022-02-01

**Authors:** Avishek Das, Gisele Cruz Camboim Raposo, Daniela Siqueira Lopes, Evair Josino da Silva, Vanda Sanderana Macêdo Carneiro, Cláudia Cristina Brainer de Oliveira Mota, Marcello Magri Amaral, Denise Maria Zezell, Renato Barbosa-Silva, Anderson Stevens Leonidas Gomes

**Affiliations:** 1Physics Department, Universidade Federal de Pernambuco, Recife 50670-901, PE, Brazil; renato.barbosa@ufpe.br (R.B.-S.); anderson.lgomes@ufpe.br (A.S.L.G.); 2Graduate Program in Dentistry, Universidade Federal de Pernambuco, Recife 50670-901, PE, Brazil; gisele.camboim@ufpe.br (G.C.C.R.); evair.josino@ufpe.br (E.J.d.S.); 3Faculty of Dentistry, Campus Arcoverde, Universidade de Pernambuco, Arcoverde 56503-146, PE, Brazil; daniela.siqueira@upe.br; 4Faculty of Dentistry, Campus Camaragibe, Universidade de Pernambuco, Camaragibe 54756-220, PE, Brazil; vanda.carneiro@upe.br; 5Faculty of Dentistry, Centro Universitário Tabosa de Almeida, Caruaru 55016-901, PE, Brazil; claudiamota@asces.edu.br; 6Scientific and Technological Institute, Universidade Brasil, Fernandópolis 15600-000, SP, Brazil; marcello.magri@universidadebrasil.edu.br; 7Center for Lasers and Applications, Instituto de Pesquisas Energéticas e Nucleares IPEN—CNEN, São Paulo 05411-000, SP, Brazil; zezell@usp.br

**Keywords:** nanomaterial, nanodentistry, optical coherence tomography, photoacoustic imaging

## Abstract

There is already a societal awareness of the growing impact of nanoscience and nanotechnology, with nanomaterials (with at least one dimension less than 100 nm) now incorporated in items as diverse as mobile phones, clothes or dentifrices. In the healthcare area, nanoparticles of biocompatible materials have already been used for cancer treatment or bioimaging enhancement. Nanotechnology in dentistry, or nanodentistry, has already found some developments in dental nanomaterials for caries management, restorative dentistry and orthodontic adhesives. In this review, we present state-of-the-art scientific development in nanodentistry with an emphasis on two imaging techniques exploiting nanomaterials: optical coherence tomography (OCT) and photoacoustic imaging (PAI). Examples will be given using OCT with nanomaterials to enhance the acquired imaging, acting as optical clearing agents for OCT. A novel application of gold nanoparticles and nanorods for imaging enhancement of incipient occlusal caries using OCT will be described. Additionally, we will highlight how the OCT technique can be properly managed to provide imaging with spatial resolution down to 10′s–100′s nm resolution. For PAI, we will describe how new nanoparticles, namely TiN, prepared by femtosecond laser ablation, can be used in nanodentistry and will show photoacoustic microscopy and tomography images for such exogenous agents.

## 1. Introduction

Nanoscience and nanotechnology, which deals with science and technology at the nanoscale, have developed in such a way that it has already been used in several areas of knowledge, with social applications and economic impact [1]. From the Internet of Nano Things [2] to health benefits [3], exploiting the nanoworld has made a big impact. This is no different in dentistry, although still in its infancy with respect to other areas of healthcare [4]. The authors of ref. [5] provided a good and updated account of nanomaterials in a diversity of dentistry applications. Among those, dental materials have received increasing attention [6], dental nanomaterials for caries management [7], restorative dentistry [8] and orthodontic adhesives [9]. Additionally, another importance to nanodentistry is silver nanoparticles, due to their antibactericidal properties [10]. Nanomaterials can be prepared, grown or synthesized using bottom-up or top town techniques, and refs. [5,11] brings an updated review highlighting recent developments regarding therapeutic applications of nanomaterials in dentistry, including chemistry, synthesis, properties and benefits of therapeutic nanomaterials over conventional materials.

From the point of view of imaging techniques, nanomaterials of different materials and shapes have already played an important role, as reviewed in [12]. Within the scope of this review, we will describe how nanomaterials have been exploited with optical coherence tomography (OCT) and photoacoustic imaging (PAI), in two modalities: photoacoustic microscopy (PAM) and photoacoustic tomography (PAT). We anticipate that both OCT and PAI are well-known and established imaging methods, as will be described in more detail in Section 3. As a common feature to both, a light source is used to illuminate (or excite) the sample, whereas the detection in the OCT is also in the optical regime, and the PAI detection is in the acoustic regime. Two important features are penetration depth and spatial resolution. For OCT, the penetration depth is limited by the absorption and scattering to a few millimeters, whereas for PAI it can achieve a centimeter range, thanks to acoustic wave propagation. Furthermore, spatial resolution, which means how small can one resolve the spatial features in the sample, can be measured in submicrometers in both methods. For OCT, this spatial resolution is determined in the axial direction by the optical source bandwidth, and the lateral resolution by the optics employed. In PAI, this is given by a combination of the incident optical geometry and acoustic detector bandwidth. Therefore, to overcome some of these issues, a combination of OCT+PAI operating in a multimodal fashion has been reported, which overcomes the penetration depth-resolution duality. Noticeable is that because the PAI detection is in the acoustic regime, a single detector can be used at any optically excited range, and this is a great advantage. This paper highlights to the readers the combination of the two imaging techniques, their application in nanodentistry which is related to the dental application in the nanoscale regime, and is organized as follows: In Section 2, we briefly give a background account on the field of nanodentistry, highlighting nanofabrication methods. In Section 3 and its subsections, we start with a general view on imaging techniques, and then describe imaging by OCT, from the basics to applications in nanodentistry. Further, we describe imaging by photoacoustics, with basic insights and imaging by PAM and PAT with TiN nanoparticles. In Section 4, we draw some conclusions and provide an outlook.

As a disclaimer, we point out that this is not a broad and deep literature review, but rather a review of some aspects of two imaging techniques (OCT and PAI) applied to nanodentistry, with literature and our own laboratory results.

## 2. Nanodentistry

Nanodentistry deals mainly with nanobiomaterials applied to dentistry and is a growing field of research with high potential for clinical translation [4,5,13,14,15,16,17,18,19]. It is a term also employed when characterization techniques in the nanoscale regime are employed for dental materials and tissues. The so-called dental nanobiomaterials are fabricated by well-developed nanofabrication processes (see more details in ref. [15]). Naturally occurring nanostructures, particularly nanobiostructures present in biotissues, are subject to intense attention from researchers worldwide, not only for imaging but also for seeing those nanostructures in situ and working is out of the scope of this article.

Nanofabrication processes are generally classified by two approaches: the top-down and the bottom-up. As the terms indicate, the former means that, by starting from bulk materials, nanosized structures are formed. Conversely, starting with atoms or molecules, nanostructures can be formed. Figure 1 shows examples of top-down and bottom-up routes, indicating some of the techniques that have been used to achieve nanostructures. From an economic point of view, the bottom-up approach is cheaper than the top-down.

As a result of the growth and development of the nanofabrication processes, it is possible nowadays to have access to organic or inorganic nanostructures, which can be dielectric, such as TiO_2_ or rare-earth-doped materials, semiconducting, as ZnO, or metallic nanoparticles, as Au or Ag, and of different structures and shapes, such as nanoparticles, nanoshells or nanorods, dimensions, and so on. These make the physical properties of these nanomaterials very morphology dependent.

Figure 2 shows examples of organic and inorganic nanomaterials, with typical shapes widely exploited in nanoscience and nanotechnology.

## 3. OCT and PAI Imaging Techniques

Diagnostics by medical imaging is one of the most prominent diagnosing techniques for pathologies due to some of their unique advantages: noninvasiveness, the amplitude of penetration depth and spatial resolution (depending upon the technique employed), real-time assessment and, nowadays, using artificial intelligence methods [20]. Besides exploiting the internet-of-things associated with 5G or the arriving 6G, this technology promotes online access for specialists anytime, anywhere [21]. Among the myriad of methods, this review focuses on two complementary and already well-known bioimaging techniques: optical coherence tomography (OCT) [22] and photoacoustic imaging (PAI) [23], focusing on nanodentistry. As already mentioned, in common, these two methods use optical excitation sources, and differ in the detection method, with OCT keeping an all-optical setup, whereas PAI uses acoustic detection. In Figure 3, a comparison of three key factors—spatial resolution, penetration depth, and imaging speed—which are characteristics of imaging techniques, were compiled by Zhu and co-workers in the context of neuroimaging [24] but applies well to other medical imaging applications. In the X-axis, the penetration depth for each technique indicated is shown, which for OCT depends upon the absorption and scattering coefficients (which, when combined, gives the tissue extinction coefficient). Note the typical 1–3 mm penetration depth for OCT, and 1mm to several cms for PAI, with an overlap region in the mm spatial regime. The Y-axis gives the spatial resolution (see Section 3.1), mainly given by the axial resolution. For OCT, it is the inverse of the source bandwidth which determines the axial resolution, such as those arising femtosecond laser sources or superluminescent diodes, or even from a swept-source with a narrow line laser. For OCT, as seen in the Y-axis of the figure, sub-micron spatial resolution can be achieved. The transverse resolution (see Section 3.1) is given by the optics employed—normally microscope objectives. On the other hand, the spatial (axial) resolution for PAI is lower than for OCT, although nowadays it can also go into the submicrometer regime (see Section 3.4 for details and references). The image speed on the right-hand side includes the computational process. For OCT, its order of magnitude varies if one considers an A, B or C-scan. For PAI, the overall image procedure is displayed on the screen for analysis. This illustrative Figure 3 shows (a) how fast OCT response is compared to PAI; (b) the penetration depth for both methods; and (c) the higher spatial resolution of OCT compared to PAI, at the expense of penetration depth. By combining both techniques for a given sample and region of interest, it may be very feasible to extend the spatial resolution range by about a factor of 1000 and penetration depth by a factor of 100. All the above features are valid for biological tissues, but can also be employed for nonbiological tissues, as long as there is absorption in the region of interest of the sample.

### 3.1. OCT: From Basics to Applications in Nanodentistry

OCT exploits the broad spectral width of a low coherence optical source, in combination with a Michelson-type interferometer, to retrieve high resolution (submicrometer) images with low penetration depth (few mm), relying on the interference between the backscattered light from tissue and the reflected beam from the reference arm. The OCT technique was pioneered in 1991 when Huang and co-workers first described a tomographic image of the eye retina [25]. Since then, scientific and technological developments in the OCT technique evolved exponentially, and the reader is referred to the authoritative book by Fujimoto and Drexler [26]. Instead of giving an extensive list of references in OCT and its diversity of applications in medical and non-medical areas, the reader is also referred to the public webpage www.octnews.org, a trusted and always updated webpage for OCT developments.

Conceptually, OCT is a very simple method (see Figure 4A): a spectrally broadband optical beam, which can be generated by coherent or incoherent sources and travels through a Michelson-interferometer, with the beam being split to the reference arm and the other arm which has the sample under study. When ultrashort Fourier transformed limited pulses are employed, they are not chirped. Otherwise, the optical sources employed are chirped. The reflected light from the surface and inner parts (including scattered light) of the biological samples thus interferes with the light returned from the reference arm, providing an interference pattern that carries the image information. The spatial resolution, Figure 4B, is provided in two directions: the axial resolution of the system is determined by the optical source and is inversely proportional to its bandwidth. The lateral resolution depends on the employed focusing optics, either lenses or microscope objectives. A very important point is the role of absorption and scattering. For this reason, an OCT system performs better in the 800 nm–1300 nm spectral window, also known as the NIR-I or NIR-II window [27]. Below 800 nm (towards the visible) absorption may hinder light penetration, and therefore working in the NIR-I and NIR-II allows deeper penetration. Furthermore, tissue scattering is also reduced as the light wavelength increases from the visible to the NIR, which is another advantage. The pioneer OCT systems operated in the so-called temporal or time-domain OCT (TD-OCT), but the technological development led to the exploitation of Fourier-domain methods, which can be spectral-domain OCT (SD-OCT) and swept-source OCT (SS-OCT). Figure 4, adapted from ref. [22], depicts the different experimental schemes for those systems, indicating the detection systems (photodetector or CCD array), axial and lateral image resolution and depth of field (which depends upon source bandwidth and optics). In the SD-OCT, because the spectrum of the beam after interference is obtained by the spectrometer, rather than by a single detector as in the SS-OCT, the nomenclature is slightly different in Figure 4A, although both are optical detectors.

The results obtained by OCT can be displayed by the so-called A-scan, whereby a single depth profile is shown (without lateral displacement of the beam); B-scan, whereby the beam is laterally displaced and the collected data is composed of a set of 2D frames and finally, the C-scan, which provides a 3D image by acquiring a volume dataset of B-scans. Figure 5 shows diagrammatically each scheme, applied to OCT in the eye as an example. For further details and alternative schemes for OCT, the reader should consult ref. [26].

### 3.2. OCT in Nanodentistry: Image Enhancement with Gold Nanoparticles

The applications of OCT in dentistry have been reviewed on a number of occasions [28,29,30]. Our group in Brazil has employed OCT for dental applications in the areas of cariology, endodontics, periodontology, aesthetics and dental materials, as exemplified by refs. [31,32,33,34,35,36,37,38,39,40,41]. OCT in nanodentistry can be seen from two perspectives: (a) the OCT technique operates such as to improve the sensitivity or detect in the nanometer regime or (b) the OCT technique can be employed to study nanostructures in dental materials or dental tissues (hard or soft).

From the first perspective, we call the reader’s attention to the work of Martin Leahy’s group [42,43,44] and the work of J. Yi and co-workers [45]. In [42,43,44], a technique called nanosensitive OCT (nsOCT) was developed and demonstrated, which exploits proper manipulation of the micrometer resolution data from Fourier-domain OCT to resolve the images in the nanoscale (10′s nm) regime and was recently applied to assess wound healing within the cornea [44]. Further, J. Yi and co-workers developed a so-called inverse spectroscopic OCT (ISOCT), which, upon quantification of the wavelength-dependent backscattering coefficient μ_b_(λ) and the scattering coefficient μ_s_,), quantify the mass-density correlation function to achieve detectable structural changes ranging from ~30 to ~450 nm. The authors validated their findings by numerical simulations, tissue phantom studies, and ex-vivo colon tissue measurements, which were cross-validated by scanning electron microscopy. For further details of both methods, the readers should assess the given references.

From the second perspective, OCT can be employed to assess nanostructures within the sample. We give some examples from our own group, whereby a commercially available OCT system was employed. In reference [46], within an international collaborative work, a pioneer application of OCT in nanodentistry was reported, whereby gold nanoparticles (AuNP) were formed in situ and immediately used as a contrast agent for dental OCT. This development was essential to imaging the desired region. Figure 6a shows an artistic view of a tooth structure, highlighting the dentinal tubules.

To perform the work, an innovative in situ photothermal reduction procedure was developed and implemented, which allowed the production of spherical AuNP inside dentinal layers and tubules. A three-step procedure was performed: First, gold ions were dispersed in the primer of a commercially available dental bonding system. Secondly, the modified adhesive system was applied to the dentin, and the dental bonding materials were photopolymerized using a commercially available photopolymerizer, simultaneously with the formation of AuNP, as shown in Figure 6b. The presence of AuNP was confirmed by scanning electron microscopy in the hybrid layer and dentinal tubules, as seen in Figure 6c. The diameter of the AuNP was determined to be in the range of 40 to 120 nm, while it is known that the dentin tubule’s diameters ranged from 500–4000 nm. Figure 6d shows the OCT images of the samples with and without AuNP. The ultra-intense regions of OCT images show highly scattering portions and high levels of direct light backscatter, due to the presence of AuNP within the imaged volume. This information is displayed in false colors in Figure 6d: warm colors for higher and cool colors for lower scattering intensities, respectively. Without AuNP, the image is not shown since it cannot be resolved. In comparison, as indicated by the arrows in Figure 6d, the tubules become detectable and easily visible with the presence of the AuNP, acting as contrast agents to enhance the OCT image. Further details can be appreciated in ref. [46].

### 3.3. OCT in Nanodentistry: Association of AuNP and AuNR as Contrast Agents for Imaging Enhancement

As another nanodentistry application, our group has exploited silver and gold nanostructures as optical clearing agents (OCA), or contrast agents, as enhancers to improve caries diagnostic by optical coherence tomography. Silver nanoparticles are well-known as biocompatible and antibactericidal agents, already employed in nanodentistry [47,48,49,50]. Nonetheless, the role of optical clearing agents in biotissue imaging is a very active subject, which found its way after Tuchin and co-workers [51], as recently reviewed in ref. [52]. A handbook on the subject covers a highly updated account of this field [53]. The final action of optical clearing agents is to provide deeper penetration depth for imaging and other healthcare purposes. As pointed out in [47,48,49,50,51,52], the physical process of optical clearing consists of the immersion/permeabilization of biocompatible chemical substances in biological tissues. Due to a series of effects of these substances on the biosample, such as the compatibility of refractive indices, cell dehydration and the increase of collagen solubility arising from osmotic properties of the OCA, the optical clearing effect occurs. OCA using biocompatible chemical substances have been used in skin, bone, cartilage, luminal organs and other connective tissues (see [52,53] and refs therein). There is a vast literature on OCAs associated with OCT applied to the skin [54], molecular diffusion in tissue [55], the role of glucose concentration [56] and articular cartilage in subchondral bone [57]. With similar aiming of enhancing imaging in OCT systems, nanomaterials—particularly gold nanoparticles, nanoprisms and nanorods, besides TiO_2_ nanoparticles—have been exploited as contrast agents in cardiovascular OCT imaging [58], angiography [59], breast carcinoma [60], as well as phantoms [61]. OCA also plays an important role in dentistry. When analyzing a tooth surface with the OCT device, the presence of caries lesions typically promotes a large increase in light scattering due to demineralization, therefore limiting the light penetration, and reducing the OCT signal before it even reaches the dentin-enamel junction [62]. In this way, OCA may provide deeper penetration and improve OCT image quality in situ, whereas an agent that modifies the diffusion properties of a sample may improve the image contrast. The use of nanoparticles as contrast agents in dentistry was already mentioned in this review [46]. More recently, silver nanoparticles (AgNP) in aqueous solution and diluted in glycerol have been exploited as OCA for a diagnostic of occlusal incipient caries lesions through OCT imaging, based on the changes in enamel birefringence and highlighting demineralized areas [47]. As a conclusion of ref. [47], OCA with AgNP evidenced the enamel birefringence and demarcated initial demineralization areas, presenting images with defined margins, a higher contrast between sound and demineralized regions, with higher OCT signal intensity in such areas. As a follow up of that work, and in order to give some more technical details, we added to this review further results using gold nanoparticles and nanorods, a biocompatible noble metal, as OCA in dentistry, whose complete account can be found elsewhere (Gisele C Cruz, MSc Thesis, “Use of gold nanostructures as optical compensation agents for incipient caries analysis with optical coherence tomography” available in Portuguese at https://repositorio.ufpe.br/handle/123456789/33796, accessed on 9 January 2022).

The experiments were carried out with ten extracted permanent molars with incipient caries lesions that were obtained from a teeth bank (Centro Universitário Tabosa de Almeida (ASCES-UNITA), Caruaru, Pernambuco, Brazil) which, after being divided into groups, were investigated for contrast enhancement by the alterations of the light extinction coefficient with OCT. Group (G1) was the control, whereas other groups were observed using the following components: glycerol (G2), AuNP (G3), AuNP diluted on glycerol (G4), AuNR (G5), and AuNR diluted on glycerol (G6). It is important to note that occlusal caries lesions are routinely detected using visual, tactile and exploratory methods in combination with radiographic exams [63], which have low sensitivity for occlusal lesions. The spherical gold nanoparticles (AuNP) were produced in the Department of Physics and Chemistry in the Universidade Federal de Pernambuco, Brazil by treating hydrogen tetrachloroaurate (HAuCl_4_) with polyvinylpyrrolidone (PVP-MW ≅ 55,000) in boiling water, where the PVP acts as a stabilizing agent. The resulting colloidal gold nanoparticles are spherical and have 2.0 ± 1.0 nm dispersed in an aqueous solution, and were characterized by the peak plasmon resonance at 550 nm [64]. The nanorods (AuNR), however, were purchased from NanopartzTM (Nanopartz Inc., Denver, CO, USA), with 50 nm diameters and 160 nm length, and were also dispersed in aqueous solution, with two plasmon resonance peaks at 520 nm and 825 nm, corresponding to the transverse and longitudinal plasmon resonances, respectively.

The studied teeth contained white or brownish spots accompanying anatomical accidents of the occlusal surfaces, but they did not have evident cavitation when analyzed by visual examination in accordance with the ICDAS code (1) first visual change in dry enamel and (2) distinct visual change in moist enamel. Teeth were excluded when caries cavities were observed by visual examination or restorations. The teeth selected for this study had their root portions included in a colorless chemically activated acrylic resin matrix with the occlusal surface parallel to the ground. Afterwards, they were placed on a PVC platform so they could be repositioned to receive the OCAs for OCT scanning. Each sample was analyzed a total of six times. The first one was the negative control (G1), which did not undergo any treatment, and then they were analyzed with five different combinations of optical clearing agents (OCA) and contrast agents (CA). Pure glycerol was a positive control (G2). The other OCA/CA combinations were: (G3) gold spherical nanoparticles (AuNP) dispersed in water; (G4) AuNP dispersed in water and diluted to 50% by weight of glycerol; (G5) nanorods (AuNR) dispersed in water; (G6) AuNR dispersed in water and diluted to 50% by weight of glycerol. After applying the OCA/CA, the specimens were scanned by OCT, and then the samples were washed with neutral detergent (Ypê, Amparo, São Paulo, Brazil) with Robinson’s brush, followed by an ultrasonic bath in deionized water for 15 min between each OCA/CA application. In this way, all samples could be scanned by OCT in the presence of each OCA/CA combination.

The OCT system employed was a commercially available model (Callisto, Thorlabs Inc., Newton, NJ, USA), operating in the spectral-domain (SD-OCT), with 930 nm of central wavelength, 100 nm of bandwidth, a maximum output power of 5 mW, resolution of 5.3/7 μm in air and water, respectively, and a lateral resolution of 8 μm and depth of light penetration of 1.7 mm. The analysis was performed along the occlusal surface, capturing two-dimensional images with an 8 mm transverse scan of the specimens. To standardize the scanning, the samples were positioned on a two-axis linear translation stage with a rotating platform and solid top plate, and images were acquired every 250 μm along the mesiodistal dimension. The two-dimensional images constitute a numerical matrix of 2000 columns at the X-axis, corresponding to maximum 8 mm scanning, and 512 rows at the Y-axis, with 1.7 mm maximum depth (in air, refractive index 1). The acquired OCT images were analyzed using in-house developed software to measure the optical attenuation coefficient (μ). The μ coefficient is a quantitative value that characterizes the decay of OCT signals when light propagates through the tissue analyzed. Thus, by measuring the μ it is possible to use the OCT signal to quantitatively discriminate between different types of tissues and their state of health [62,65]. For the μ coefficient analysis, the software operator uploaded the images of each studied group. In each image, the operator selected the width and position of two windows with the region of interest (ROI). For standardization, the window width was maintained constant (10 A-scan) during all the analyzes. As each biological sample presents distinct morphological characteristics, the position of the windows was adjusted for each image, always positioning one window in a region of healthy tissue and the other one in a region of abnormal tissue.

Using the B-scan mode, the software identifies the surface of the tissue (air-tissue interface) and uses it as a reference to perform an average of the normalized A-scan inside each ROI window. The A-scan was normalized by the maximum value to avoid erroneous values due to possible reflection on the first surface. The average A-scan of each ROI window was used to fit an exponential decay, based on the Beer–Lambert law, I(Z) = I_0_ exp (−μz), where I(Z) is the intensity as a function of depth, z and μ is the optical attenuation coefficient. The obtained μ value was recorded for posterior analysis. This procedure was repeated for each image. Statistical analysis was performed using GraphPad Prism 7 software (GraphPad Software, Inc., San Diego, CA, USA). A mean and standard deviation of each group were calculated. Normal distribution was not performed by the Kolmogorov–Smirnov test. To analyze differences between the groups, the Kruskal–Wallis test was used. In all groups, Student’s *t*-test was performed. To verify if there was a difference between groups, the “repeated measures of variance RM-ANOVA” test was used. The statistical significance of all tests should be considered as *p* < 0.05.

In order to quantify the OCT signal intensity in the dentin-enamel junction (DEJ) area, the μ coefficient of each group was obtained in sound and affected areas (Table 1). Figure 7 shows the B-scan (2D pictures) and the A-scan graphics (on the right-hand side) of the window corresponding to the pit area in G1 (control), G3 (AuNP), and G6 (AuNR + glycerol). Both G3 and G6 evidenced the greatest difference in the attenuation coefficient compared with the control group. The images were obtained with a horizontal sweep of 6mm, and the bar scale is the same for all pictures. The central yellow line shows where the A-scan was measured.

An inspection of Table 1 shows an increase in the light attenuation in all carious tested groups when compared with G1 (0.184). It can be observed that groups three and six obtained a higher μ (0.224) on the carious surface. However, at the sound areas, gold nanorods diluted in glycerol (G6 = 0.168) showed an attenuation coefficient similar to that obtained from the isolated samples in the presence of glycerol isolated G2 (0.169).

As already discussed in this review, OCAs may enhance the OCT depth penetration, whereas a contrast agent that modifies the diffusion properties of a sample may improve the image contrast. The results of this section, exploiting gold nanoparticles and gold nanorods, confirmed that the use of contrast agents increased the optical attenuation coefficient of subsurface lesions located under enamel on the occlusal surface. Furthermore, the difference between the optical attenuation coefficients of sound teeth and carious lesions increased, making the gold NP and NR a potential tool to improve the identification of lesions. Glycerol is one of the most used and tested OCAs, both in soft and hard tissue, as already indicated by several references. It has a high viscosity, 1.42 pa at room temperature, and refractive index n = 1.47. This is well suited for use in dental imaging due to its biocompatibility, constituting an important advantage for OCT imaging, and therefore justifying the use of glycerol in some groups (G2, G4 and G6) of the present study. Within the spectral range of interest in this study, the attenuation coefficient remained similar in all groups which had a glycerol addition, even with gold nanoparticles.

In order to improve the NP OCAs properties, such as viscosity, the dispersion of the NP in glycerol was tested (to compare with them in water, the vehicle used to NPs which is less viscous than glycerol). However, the results, in this case, did not show a statistically significant difference. Finally, we note that, in contrast with the literature, the glycerol increased the optical attenuation coefficient, which means that the light penetrated less than in the control group, and thus opposed literature findings, as this agent promoted an increase in light penetration [55,66]. One possible explanation is that in the mentioned refs. [55,66], the OCA is used in soft tissues, not hard tissue as here; however, this claim needs to be further verified.

### 3.4. PAI: From Basics to Nanodentistry Applications

As pointed out in the introduction of this section, PAI relies on the acoustic response of a biotissue upon excitation with light, which is absorbed by the medium. The absorbed photons lead to local heating, followed by the expansion/contraction and generation of acoustic waves. PAI can be understood as a hybrid modality that relies on the interplay between absorption of light with subsequent ultrasound emission. It is a well-known imaging modality [23,67,68] and well-developed technologically.

Figure 8 shows the basic principles of the photoacoustic effect [69]. A pulsed optical source, generally a laser due to its high intensity, is used to excite the medium under study. Once the light is absorbed by endogenous or exogenous substances, the target is heated, which in turn generates acoustic waves. This created photoacoustic signal is directly proportional to the light absorption and excitation fluence, (fluences of ~mJ/cm^2^, below tissue damage threshold, are used, and no damage was verified to the tissue), and is not affected by light scattering, since the acoustic scattering signal of tissues is approximately three orders of magnitude less than optical scattering. That fact leads to one of the main advantages of PAI in relation to all-optical imaging methods, which is its ability for deeper imaging (several cms) while a spatial resolution of ~100 μm is kept [67].

PAI has emerged in different modalities, as shown in Table 2, from ref. [70]. The different methods for PAI can be implemented with proper modifications from the basic experimental setup. As seen in Table 2, clinical applications of PAI are already being employed. PAI has already been demonstrated for imaging at a molecular or cellular level, as seen in Figure 9, which shows its multiscale capability, particularly photoacoustic computed tomography (PACT), which can be exploited from a single cell or organelles to whole-body small animal dimensions.

Whereas endogenous substances are widely used for light absorption, exogenous contrast agents for PAI also play an important role in the image generation from a targeted biological site. Although PAI already provides deeper penetration depth in tissue, wavelengths in NIR-II and the NIR-III biological window will gain further benefits. 

The scope of PAI can be significantly expanded, for instance, to several cms if deeper tissue-penetrating light in the NIR-II window could be used to excite an exogenous PA contrast agent. In this sense, a recent development is nonlinear PAI using multiphoton absorption (two or more photons) of a NIR light which can be used as a mechanism to overcome or mitigate this challenge [72]. In ref. [73], the photoacoustic Z-scan was employed to estimate the efficiency of photothermal relaxation, therefore characterizing the metrics for a photoacoustic contrast agent to be used for nonlinear photoacoustic imaging. Besides employing endogenous substances or biocompatible exogenous as contrast agents for PAI, nanoparticles as gold−silica nanoshells [74] or conjugated polymer [75] nanoparticles have also been used as a contrast agent for photoacoustic tomography. The applications of PAI in dentistry started in 2006 [76] and have since been used in studies of caries diagnostics, periodontology, dental implants and blood detection in dental pulp [77,78,79,80,81,82]. Due to the already described difficulties for imaging approaches to detect occlusal caries, besides OCT with the aid of OCA described before, we employed PAI, and more specifically PAM, to image occlusal caries [83]. The results of ref. [83] are shown in Figure 10. Part A shows the experimental setup, which employed a Q-switched Nd:YAG laser with selective wavelengths of 532 nm and 1064 nm, a pulse width of 6 ns, and a repetition rate of 10 Hz. The laser beam diameter was measured to be approximately 7 mm. The sample (extracted human tooth) was placed on a holder inside a distilled water tank. The entire assembly was supported on a computer-controlled motorized X-Y translation stage. The PA signals were acquired by employing an immersion-type ultrasonic transducer V310-N-SU (Olympus, Waltham, MA, USA) with a center frequency of 5 MHz. Part B of the figure shows the PA imaging results for excitation at 532 nm and 1064 nm for a sound tooth, as well as teeth with incipient and advanced caries. The transducer was partially immersed into the water pointing at the laser beam-tooth interface at 45° for optimal acoustic coupling with the sample. For the depth measurements shown in part C, the PA detector was placed on the tooth side and moved vertically from top to bottom with a computer-controlled Y-axis translation stage. For the measurements, the light fluence was kept at 17 mJ/cm^2^, and therefore below the ANSI established threshold. Besides clearly seeing occlusal caries, it is worth noting that, at either wavelength, images with a penetration depth of 4 mm could be detected—well above the noise floor shown in part C of Figure 10. Further development of the setup to perform PAT shall provide further insight into the tooth structure, thus enabling diagnostics of other features.

PAI exploits different types of nanomaterials, and in ref. [70] the readers can find a very good account of PAI being applied to nanomedicine. Among the photoacoustic agents based on nanoparticles, we point out a somewhat unexploited metallic nanoparticle, Titanium Nitride (TiN), which is a promising nanomaterial for biomedical applications, including PAI [84]. TiN is an alternative plasmonic material to other noble metals, such as gold or silver, presenting a redshifted plasmonic extinction band (600–800 nm), well within the first biological window. As we already highlighted, one possible way to improve further depth penetration of light in biotissues is to use wavelengths in the NIR, and benefit from multiphoton absorption. We have recently characterized the nonlinear photoacoustic response of TiN NP using a photoacoustic Z-scan [85]. The TiN NPs in solution was synthesized by femtosecond laser ablation, starting from a solid target, as detailed in ref. [84]. Figure 11 and Figure 12 show the details of the experimental setup, morphological characterization and nonlinear absorption measurements for TiN NP.

As the results from the measurements and theoretical fits (see ref. [85]), the values of nonlinear absorption coefficients from PAZ-scan (and optical Z-scan, between parenthesis) are obtained to be 3.27 ± 0.17 × 10^−8^ (2.40 × 10^−8^), 6.41 ± 0.32 × 10^−8^ (1.04 × 10^−8^), and 3.22 ± 0.16 × 10^−8^ (1.23 × 10^−8^) cm/W for an excitation wavelength of 600 nm, 700 nm, and 800 nm, respectively.

The studies of ref. [85] corroborated the importance of TiN NP for applications where linear and nonlinear absorption plays an important role. Although NP has been used as agents for PAI in a diversity of applications, to the best of our knowledge, applications in nanodentistry have not been reported. TiN may be an important NP for this application, and we present for the first time bidimensional PAM in TiN NP solution in acetone, placed inside an Eppendorf tube. The experimental scheme and setup were similar to Figure 10A, and Figure 13 and Figure 14 show the obtained images. In Figure 13, the proof of concept for the PAM is shown.

The lateral resolution in the PAM system is determined by the diffraction-limited optical spot size [86,87], which is usually limited by the beam size of 1000 μm. The axial resolution of the PAM (*AR_PAM_*) is related to the velocity of sound in the coupling medium and bandwidth of the ultrasonic transducer as [86,87]:(1)$$AR_{PAM}=0.88\frac{v}{\Delta f_{c}}$$
where *v* is the speed of sound in the medium and $\Delta f_{c}$ is the bandwidth of the PA signal which is approximately equal to the ultrasonic transducer bandwidth. Considering, the speed of sound of 1160.96 m/s in acetone media (since TiN NPs are suspended in acetone) [88] and a −6 dB detection bandwidth of 83.84%, the axial resolution is calculated to be 244 μm. Such lateral and axial resolution can further be enhanced to subwavelength by employing high numerical aperture objective optics and high bandwidth transducers [86,89]. The total time for a full X-Y PAM scan is observed to be around 20–25 min. In Figure 14, we show a very promising result of three-dimensional (3D) PAT imaging of TiN NP in both bare and beneath tissue form.

The Eppendorf tube with the TiN NP was placed underneath 2 cm thick chicken breast tissue, and 2D-PAT imaging was clearly obtained for each vertical position (see Figure 14d). To obtain a 3D-PAT image, the 2D-PAT slices obtained in Figure 14c,d are rendered by Image J software, as shown in Figure 14e,f.

Although we employed only two wavelengths in our demonstration studies—532 nm and 1064 nm—a tunable source, such as an optical parametric oscillator can be used from the visible to the NIR in PAI systems. If there is enough endogenous absorption in specific wavelengths, or, if exogenous absorbers are to be used, they can be tailored to absorb in required regions. This is one of the material’s scientists’ important tasks, and nanomaterials are particularly important to be tailored for specific absorption regions.

## 4. Discussion and Conclusions

Nanodentistry is a young and growing area of research, with an open road ahead, both from the scientific as well as from the technological and clinical viewpoint. This review highlighted several aspects of nanodentistry, with literature examples and some novel results using two different imaging methods, OCT and PAI. With OCT, applications in nanodentistry have already been reported, and several examples were given. Additionally, original results on the use of gold nanostructures—nanoparticles and nanorods—to improve OCT contrast with OCA. The presented results demonstrated the potential of gold nanostructures to improve imaging by OCT, and further studies and applications can be envisaged. We also highlighted and referenced an important method, called nanosensitive OCT, which is a potentially applicable method to nanodentistry. One aspect related to the use of nanomaterials in biotissues, particularly in vivo, is related to nanotoxicity. This is still debatable and there is a vast literature on the subject, as can be seen from ref. [6]. One important point is the concentration of nanoparticles, besides, of course, its material, which should be within a range already accepted and demonstrated not to be harmful [6]. Regarding PAI, two modalities, PAM and PAT, were described, and again these methods are widely known and exploited, even with nanomaterials as exogenous agents. Applications in biotissues, including soft and hard tissue from the oral cavity, have been reported. One important issue to be addressed is the use of PAI with heterogeneous materials. The literature on this subject is still scarce [90,91] and it certainly deserves further studies, particularly when applied to dentistry and nanodentistry. However, PAI applied to nanodentistry, to the best of our knowledge, has yet to be reported. We described a nanomaterial, TiN, which has several advantages for PAI, such as easy fabrication when using femtosecond lasers, biocompatibility and plasmonic enhancement. We have shown its nonlinear optical characterization using a photoacoustic nonlinear method, and also two examples of PAM and PAT images, with a 3D image in the latter case.

Looking ahead, further studies and research in nanodentistry can find important applications in detecting biofilm formation in natural or artificial teeth, which is the starting point for caries or other oral diseases in the case of artificial teeth, particularly in complete or partial dentures. Another way where nanomaterials are important in nanodentistry is in dental materials. This requires new developments, which include rare-earth-doped nanomaterials, which can be simultaneously used for imaging, temperature measurements and curing, using, for instance, photodynamic therapy [92]. Antibacterial nanotherapy is also an interesting field of research. Periodontal diseases can also benefit from nanodentistry, exploiting the so-called nanorobots. An extensive list of different aspects of novel approaches in nanodentistry can be found in ref. [93].

## Figures and Tables

**Figure 1 nanomaterials-12-00506-f001:**
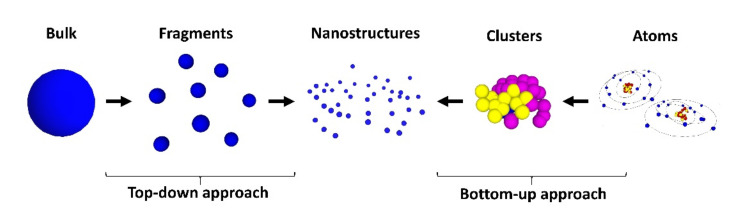
Top-down and bottom-up approaches used in nanofabrication.

**Figure 2 nanomaterials-12-00506-f002:**
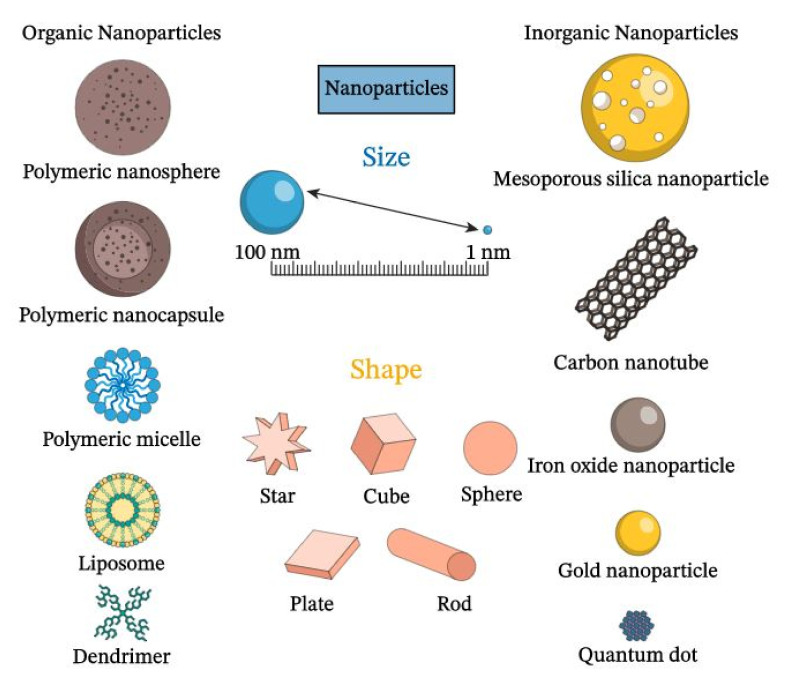
Types and shapes of nanoparticles. (Downloaded from https://www.nagwa.com/en/explainers/640142370207/, accessed on 7 November 2021).

**Figure 3 nanomaterials-12-00506-f003:**
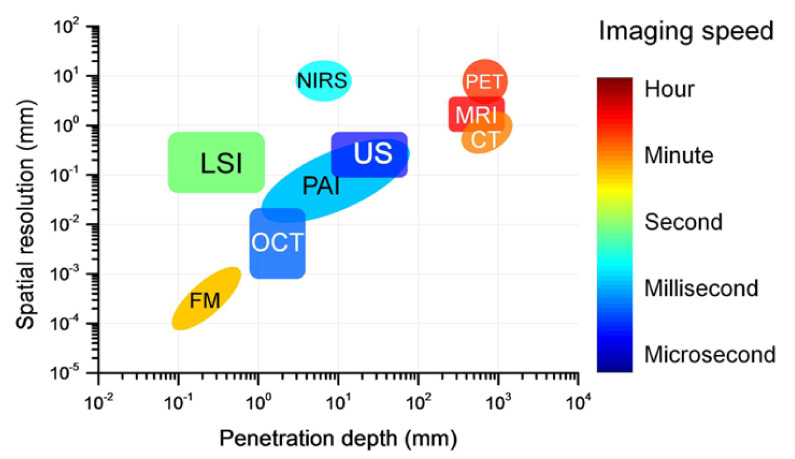
Comparisons of the penetration depth, spatial resolution and image speed among various techniques for neuroimaging, including magnetic resonance imaging (MRI), computed tomography (CT), positron emission tomography (PET), ultrasound imaging (US), optical coherence tomography (OCT), photoacoustic imaging (PAI), fluorescence microscopy (FM), near-infrared spectral imaging (NIRS), and laser speckle imaging (LSI). Reprinted with permission from ref. [24].

**Figure 4 nanomaterials-12-00506-f004:**
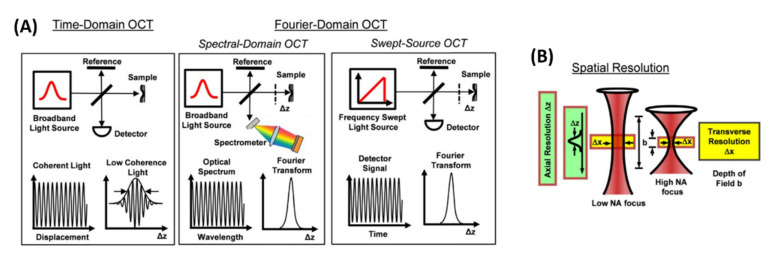
(**A**) Key concepts of OCT, including TD-OCT, SD-OCT, and SS-OCT detection; (**B**) Axial and lateral image resolution and depth of field; Adapted with permission from ref. [22].

**Figure 5 nanomaterials-12-00506-f005:**
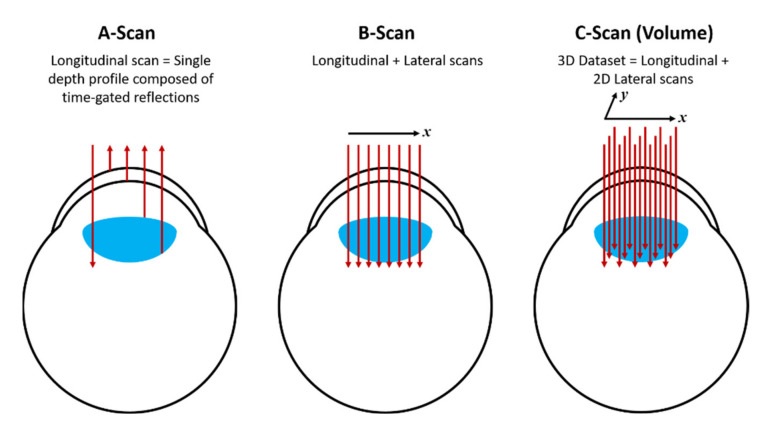
Diagram showing the A-scan, B-scan and C-scan obtained from the OCT method.

**Figure 6 nanomaterials-12-00506-f006:**
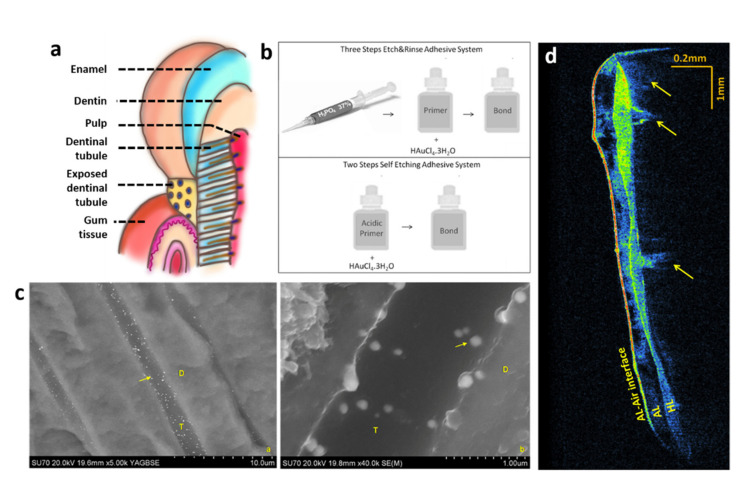
(**a**) Artistic rendering of dentinal tubules and its connection to the pulp; (**b**) Steps for the in situ gold nanoparticle preparation; (**c**) SEM images of dentinal tubules, showing the gold NP; (**d**) OCT image of the dentinal tubules, only possible due to the presence of gold NPs. Figures (**b**–**d**) reprinted with permission from [46].

**Figure 7 nanomaterials-12-00506-f007:**
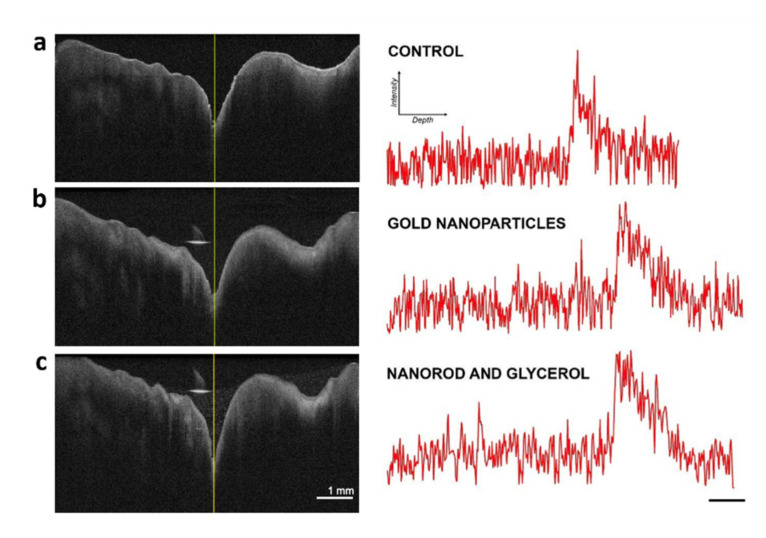
OCT B-scan images across occlusal surface of a molar, followed at right by the corresponding light intensity decay obtained from an A-scan of the region of interest (ROI) in the central pit area, according to OCAs used on the surface: (**a**) G1—Control; (**b**) G3—AuNPs; and (**c**) G6—AuNRs in glycerol.

**Figure 8 nanomaterials-12-00506-f008:**
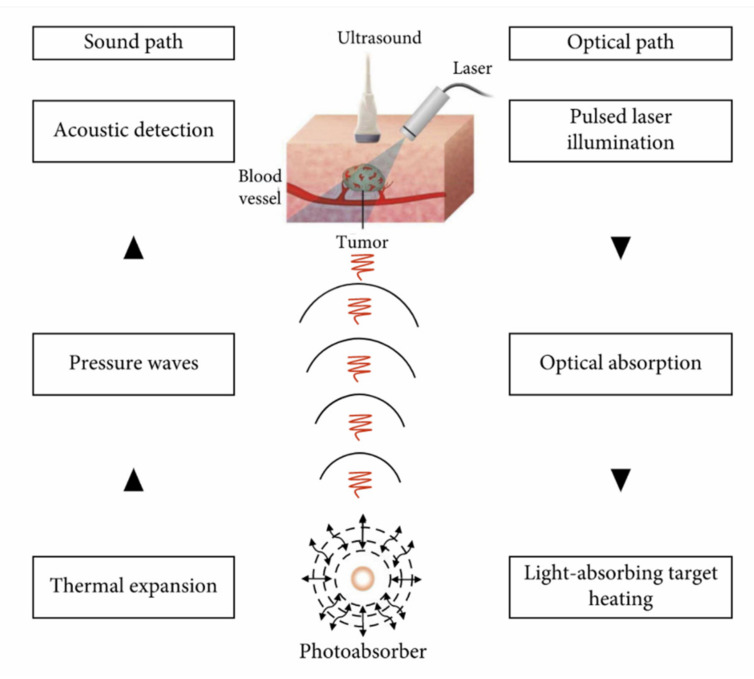
Physical principles of PAI. Figure reprinted with permission from [69].

**Figure 9 nanomaterials-12-00506-f009:**
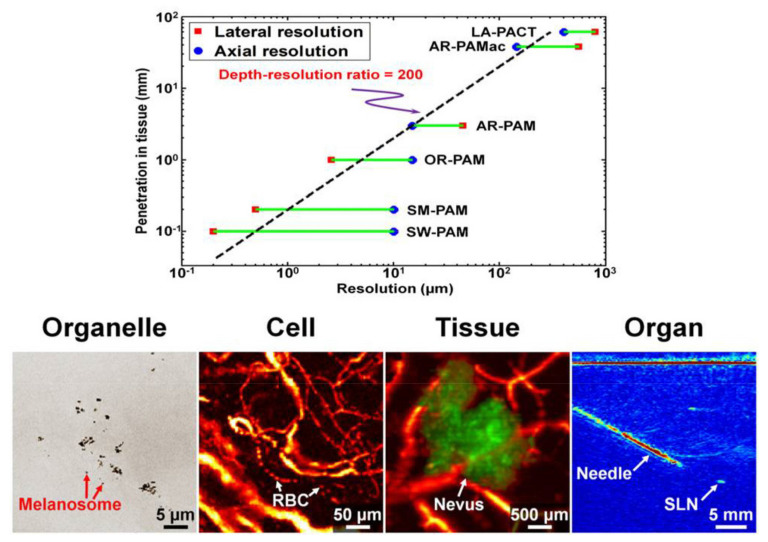
The multiscale capability of PAI. Figure reprinted with permission from [71].

**Figure 10 nanomaterials-12-00506-f010:**
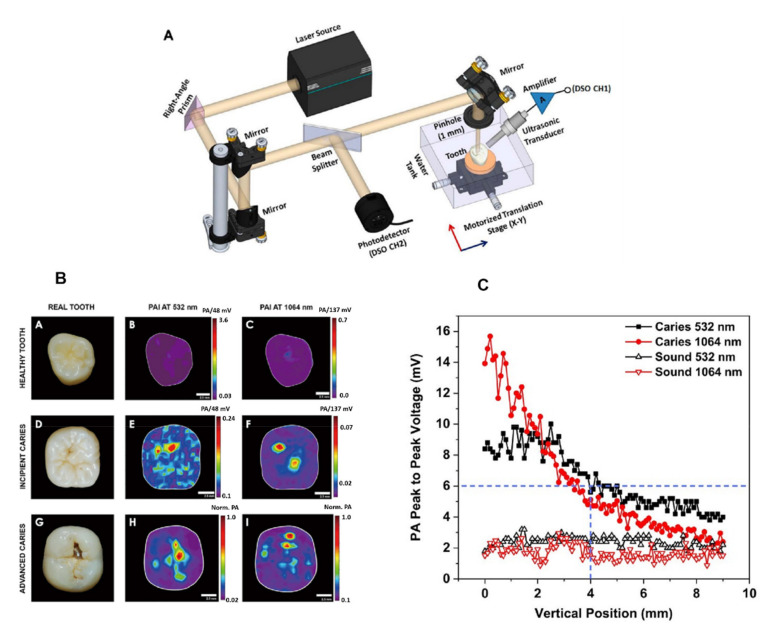
(**A**) Schematic diagram of the lab-made photoacoustic imaging system. (**B**) Photoacoustic (PA) images of sound and carious teeth at 532 nm and 1064 nm with a 5 MHz photoacoustic detector. The rows represent the sound tooth (**A**–**C**); incipient caries (**D**–**F**); and advanced caries (**G**–**I**) groups. From the left to the right, the first column shows the photographic image of the representative samples, and the second and third columns show the corresponding PA images at 532 nm and 1064 nm, respectively. PAI: photoacoustic imaging. (**C**) Peak to peak voltage for the depth of the photoacoustic signal in a tooth with incipient caries. The scale bar is 2.5 mm long. From ref. [83] with permission.

**Figure 11 nanomaterials-12-00506-f011:**
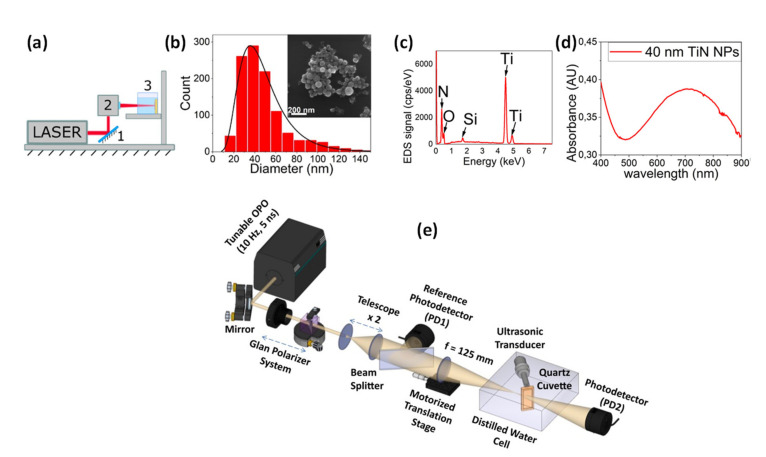
(**a**) Schematic of the fs laser ablation method; (**b**) Size distribution of TiN NP prepared by laser ablation; (**c**) EDS of the NP, identifying the NP contents; (**d**) Absorbance spectra of 40 nm TiN NP in acetone; (**e**) Experimental scheme for photoacoustic Z-scan. Reprinted with permission from ref. [85]. Copyright Optica Publishing Group, 2020.

**Figure 12 nanomaterials-12-00506-f012:**
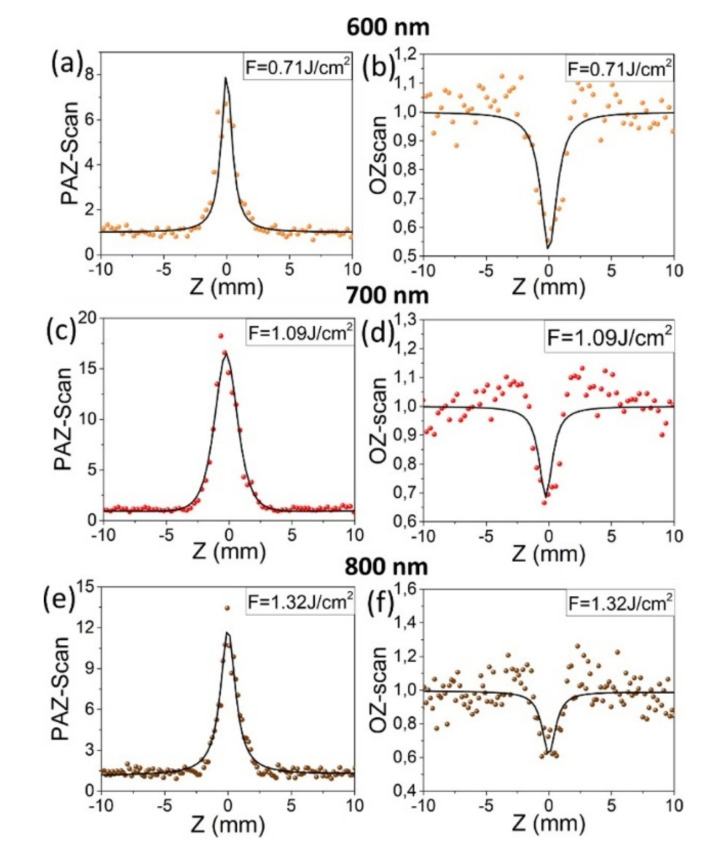
Experimental results (dots) and theoretical fits (lines) of the nonlinear absorption from TiN NP with the OPAZ setup. Left column (**a**–**c**), PAZ-scan measurements; right column (**d**–**f**), optical Z-scan with open aperture, simultaneously measured. Reprinted with permission from ref. [85]. Copyright Optica Publishing Group, 2020.

**Figure 13 nanomaterials-12-00506-f013:**
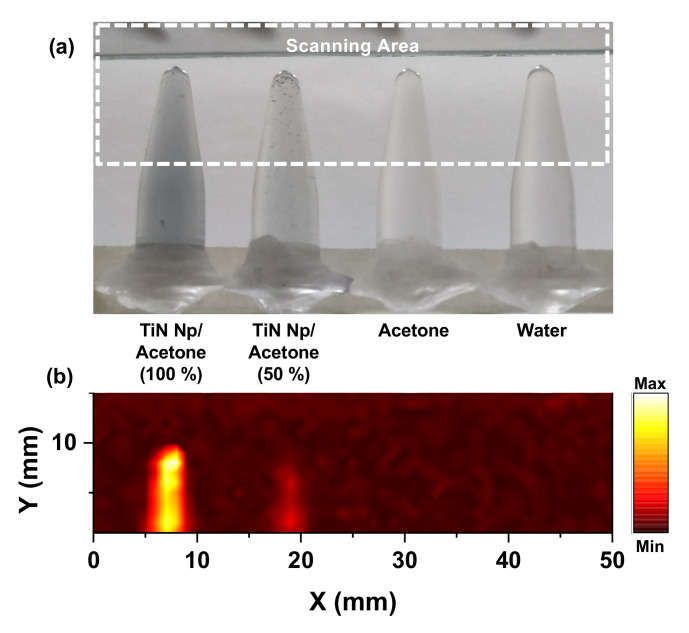
(**a**) Image of TiN NP/ acetone samples with 100% and 50% concentration, acetone and D.I. water, (**b**) PAI of the samples excited with 6 mJ/cm^2^, 532 nm, 10 Hz, ~5 ns pulsed laser source. The PAI is obtained with a single element 5 MHz ultrasonic transducer, (V310-N-SU (Olympus, Waltham, MA, USA)).

**Figure 14 nanomaterials-12-00506-f014:**
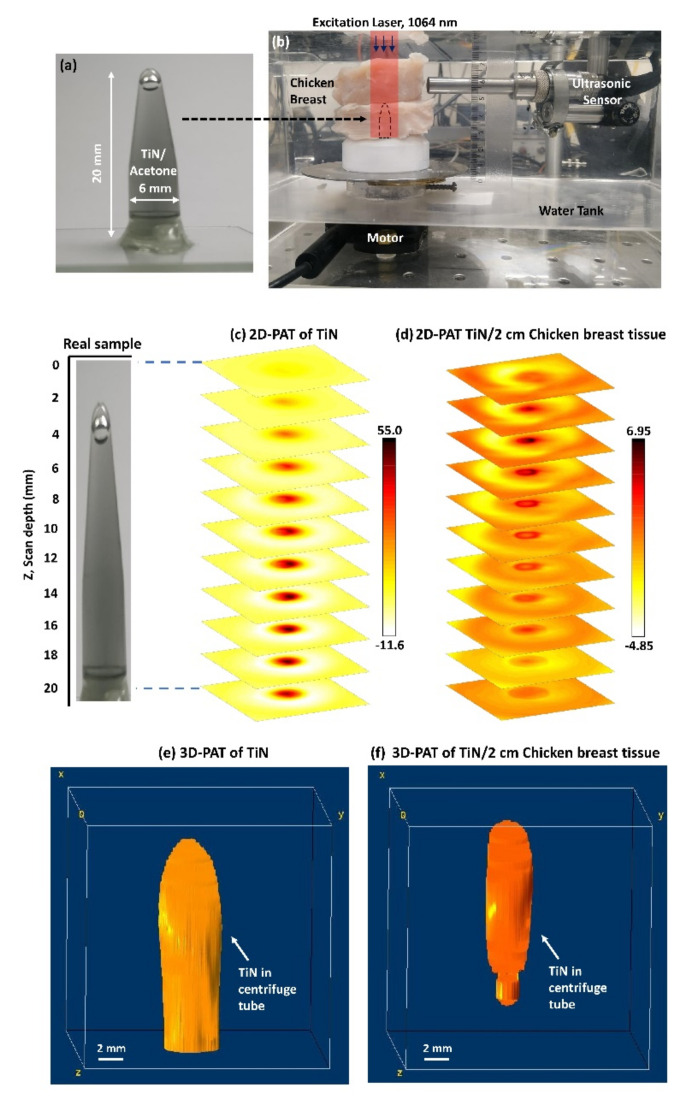
(**a**) Real picture of TiN/acetone sample filled inside a centrifuge tube (length = 20 mm, OD = 6 mm), (**b**) 3D-PAT setup containing TiN sample covered with 2 cm of chicken breast tissue. The PAT is performed for both bare TiN and TiN with chicken under 5 ns, 10 Hz, 1064 nm excitation where the peak energy density on the sample surface is maintained to be 40 mJ/cm^2^; depth-resolved 2D-PAT reconstructed image slices for (**c**) TiN and (**d**) TiN covered with 2 cm thick chicken breast tissue, obtained from a different scan depth. Every slice dimension is 20 mm × 20 mm; 3D-PAT reconstructed image for (**e**) TiN and (**f**) TiN covered with 2 cm thick chicken breast tissue, obtained after rendering 2D slices (from (**c**,**d**)) in Image J software. The axes are scaled to 20 mm × 20 mm × 20 mm.

**Table 1 nanomaterials-12-00506-t001:** Mean ± S.D. of the μ coefficient (mm^−1^) values, taken from OCT images of sound and carious areas of teeth analyzed with optical clearing agents and contrast agents (OCA/CA). The RM-ANOVA analysis indicates that the values are not statistically different in the groups for sound and caries surfaces (*p* ˃ 0.05).

Surface	Group (G)	N	Mean (mm^−1^)	SD	Minimum	Maximum	*p*-Value ¹
**Sound area**	Control (G1)	10	0.161	0.04	0.093	0,251	<0.0001
	Glycerol (G2)	10	0.169	0.06	0.087	0.257	<0.0001
	Gold Nanoparticles (G3)	10	0.133	0.06	0.036	0.25	0.0001
	Gold Nanoparticles and Glycerol (G4)	10	0.143	0.05	0.063	0.242	<0.0001
	Nanorod (G5)	10	0.135	0.07	0.007	0.233	0.0002
	Nanorod and glycerol (G6)	10	0.168	0.04	0.086	0.248	<0.0001
**Carious area**	Control (G1)	10	0.184	0.05	0.136	0.292	<0.0001
	Glycerol (G2)	10	0.198	0.07	0.069	0.307	<0.0001
	Gold Nanoparticles (G3)	10	0.224	0.03	0.145	0.276	<0.0001
	Gold Nanoparticles and Glycerol (G4)	10	0.191	0.03	0.116	0.259	<0.0001
	Nanorod (G5)	10	0.214	0.04	0.158	0.269	<0.0001
	Nanorod and glycerol (G6)	10	0.224	0.05	0.109	0.291	<0.0001

¹ One sample *t*-test.

**Table 2 nanomaterials-12-00506-t002:** Different modalities of PAI. Table reprinted with permission from [70].

Modality	Application	Advantage	Status
PACT	Peripheral joints, brain, whole-body study	Real-time and tomographic imaging	Preclinical and clinical
PAM	Molecular or cellular imaging	Microcirculation imaging without exogenous contrast	Preclinical
PAE	Gastrointestinal or cardiovascular imaging	Gastrointestinal tract imaging	Partially clinical
PAFC	Circulating tumor cells detection	Quantitative flow cytometry imaging	Preclinical
mmPAI	Circulating tumor cells detection	Specific contrast enhancement availability	Preclinical

Abbreviations: PAI, photoacoustic imaging; PACT, photoacoustic computed tomography; PAM, photoacoustic microscopy; PAE, photoacoustic endoscopy; PAFC, photoacoustic flow cytometry; mmPAI, magnetomotive PAI.

## Data Availability

Data from the authors publications reviewed here are available upon request to the corresponding author.

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
