# Peer review of "Exploiting Nanomaterials for Optical Coherence Tomography and Photoacoustic Imaging in Nanodentistry"

_nanomaterials, 2022, doi:10.3390/nano12030506_

Round 1
Reviewer 1 Report
Das et al present in their paper a quite interesting review paper on using OCT and PA imaging in nano dentistry.
Understanding and of course improving the performances and capabilities of the multimodality imaging systems is a hot topic in the research community, and therefore could be of high interest, provided that substantially new and interesting information is presented.
The manuscript presented to me for reviewing, as the authors mention in the abstract is supposed to be a review on the state-of-the-art PA/OCT capabilities, but at the same time, not very usual to me, they introduce new concepts and show new findings which should, in my opinion, be the subject of another journal paper. Instead of having a nice review paper, the reader is going to face a complex paper with information sometimes chaotically presented.
Overall, the authors show a quite interesting work, which has got some potential, but in its present form, as it lacks clarity and careful consideration of all the details involved by the 2 technologies, I am not quite sure it can be published without some additions and rectifications.
In the event the paper is deemed for publication I suggest the authors of the paper try to address the following aspects:
- The motivation of the paper does not seem to be properly emphasized. Both technologies, PA and OCT are well established/demonstrated imaging techniques, but what is the advantage of considering them over others? Advantages and limitations of these 2 modalities that should be presented from the real beginning; can some limitations be overcome by a multimodality imaging instrument PA+OCT? All these should be presented in the Introduction, so the reader has a clear idea about the motivation the authors have before going into the core of the paper. Some clarifications should be also added on how imaging technics providing resolutions in the um range are useful to look at nanostructures.
- In Fig 3, along the vertical axis, I suppose we have the axial resolution. The authors have to clarify all over the paper what is the meaning of the spatial resolution. What would be interesting for the authors to present is a similar plot where along the vertical we have lateral resolution and the imaging speed to refer to how quickly a transversal image is produced. In Fig actually, it is not clear what we have. Is the speed of acquiring data to produce an A-scan, or B-scan, or C-scan, or the time to process data is also included? Penetration depth refers to what samples? It is probably the biological tissue, but it must be specified.
- In Fig 4, what is the difference between the Optical spectrum and Detector signal? To me, it is the same. The spectra shown should be chirped unless they are pre-processed, in which case the authors should mention it. The process of building up images is clear to those in the field, but not useful to anyone else. I suggest the authors at least add a sketch of the eye and present proper definitions for the A-scans, B-scans, C-scans. Please add labels for the objective. Not clear what is shown in the picture with delta z as there are no labels.
- Some images in Fig 5 are of very poor quality. All sub-images should be labelled.
- Not sure why there is a need to show an optical polarization microscopy image in Fig 6 in a paper on OCT and PA? It makes the paper far too complex. What are the characteristics of the images shown in Fig 6? Sizes, resolutions, etc?
- Same, for 7, no scale bars. Please show where the A-scans presented on the side were collected from. Add scales as well.
- In using photo-acoustic, the power delivered to the sample must be controlled in such a way that the sample is not damaged. When it comes to using photo-acoustic to image teeth, are the authors aware of any work on the limitations of the energy per pulse in dentistry? Here, typically ns pulses duration are used. Can’t micro-cracks appear in the tooth?
- Small fonts, low-resolution images in Fig. 10. Image A cannot help the reader unless is presented in a clearer way.
- Can the authors also comment on the limitations of PA when imaging heterogeneous materials and present a couple of papers showing some work on this issue with reference to nano dentistry?
- The authors only mention PA instruments operating at 532 or 1064 nm. What about other wavelengths? What is the optimum wavelength to be used in dentistry? What about some spectral PA analysis? The nano-structure can be tuned to absorb light at other wavelengths.
I hope that by answering some of my comments the authors will manage to improve their paper.
Reviewer 2 Report
Review of Nanomaterials-1508243
Title: Exploiting nanomaterials for optical coherence tomography and photoacoustic imaging in nanodentistry
Authors: A. Das et al.
This review paper covered the overall applications of nano-sized contrast agents made of gold or organic compounds to improve OCT and Photoacoustic images in terms of contrast or penetration depth in dentistry. To my view point, the manuscript is well-written in English and explained the many examples and results of nanomaterial assisted contrast enhancements in OCT and PAT/PAM imaging of human teeth in vivo or ex vivo as well. Hence, I think that the paper could be accepted with correction of following minor comments.
- At line 206 - “Fig. 5d show the OCT images of the samples with and without AuNP.”, there is just one image in Fig. 5d. Please show comparison of OCT images of same dental tubule before and after applying the AuNPs.
- Please discuss about the effect of accumulated nanostructures in teeth such as toxicity or harmfulness in our body.
